applied mathematics/statistics

automation, network science, occupational skill, job transitions

**Author for correspondence:**
Jordan D. Dworkin
e-mail: jdwor@pennmedicine.upenn.edu

# Network-driven differences in mobility and optimal transitions among automatable jobs

Jordan D. Dworkin

Department of Biostatistics, Epidemiology, and Informatics, Perelman School of Medicine, University of Pennsylvania, Philadelphia, PA, USA

JDD, 0000-0002-5720-1298

The potential for widespread job automation has become an important topic of discussion in recent years, and it is thought that many American workers may need to learn new skills or transition to new jobs to maintain stable positions in the workforce. Because workers' existing skills may make such transitions more or less difficult, the likelihood of a given job being automated only tells part of the story. As such, this study uses network science and statistics to investigate the links between jobs that arise from their necessary skills, knowledge and abilities. The resulting network structure is found to enhance the burden of automation within some sectors while lessening the burden in others. Additionally, a model is proposed for quantifying the expected benefit of specific job transitions. Its optimization reveals that the consideration of shared skills yields better transition recommendations than automatability and job growth alone. Finally, the potential benefit of increasing individual skills is quantified, with respect to facilitating both job transitions and within-occupation skill redefinition. Broadly, this study presents a framework for measuring the links between jobs and demonstrates the importance of these links for understanding the complex effects of automation.

## 1. Introduction

There are few topics that have garnered as much concern among the political, academic and working classes around the United States as the potential automation of millions of jobs [1–3]. Although the scale of disruption that will result from this automation is debated [4–6], the possibility of widespread effects has led to much public discussion and worker anxiety. In recent years, policy-makers have worked to develop

governmental solutions, companies have sought to train workers for the new job market, and non-profits have attempted to re-tool new and old workers for future success.

To supplement and better inform these vital efforts, researchers have begun the task of performing rigorous quantitative studies on the potential effects of automation. This work has been carried out at the level of nations, cities, job sectors and individual jobs [7–11]. To allow for more fine-grained research on this topic, widely cited work [12] presented a method for estimating the susceptibility of hundreds of individual jobs to computerization. Recent studies have then built upon this work to investigate the potential differential effects of automation for different regions of the United States [8], and even to question its possible political impacts [13]. Yet while previous studies have grouped jobs by their regional presence or broad sectors, the effect of automation has generally been investigated with the assumption that a job's outlook is related only to its own automatability.

While this assumption allows for a valid study of certain characteristics of automation, it may obscure the ways in which automation affects not only the jobs that workers currently hold, but the ability of those workers to transition into other, less automatable jobs. Job transitions are likely to be an important feature of the next few decades of work, with recent studies estimating that 54 million US workers may need to learn new skills and switch careers by 2030 [6]. Yet while programmes exist to facilitate job transitions, several obstacles stand in the way of their success. Specifically, programmes have been plagued by a lack of funding [14], procrastination and false expectations among workers [15], gendered stigmas about growing jobs [16] and worker reliance on social security and disability benefits [17].

Because of the many difficulties in facilitating effective job transitions, it is important to gain a better understanding of how workers can best navigate the job market, which workers in at-risk jobs may need the most assistance from third parties in making successful transitions, and what transitions should be recommended based on workers' positions in the labour market. As prior research has demonstrated that shared necessary skills are related to workers' ability to transition between jobs [18], the use of skill overlap provides an opportunity to examine more integrated and complex effects of automation.

In this study, I operationalize the job market as a network, in which jobs are more or less similar to each other based on the skills, knowledge and abilities that they require. Using this framework, I seek to answer three main questions. First, does the structure of the network affect the mobility of some at-risk workers more than others? Second, to what extent should transition recommendations take jobs' similarity into account? Third, which skills represent potential opportunities for retraining? In the next section, I describe the data and methods used to address these questions. I then present the results of these analyses and discuss the interpretations and potential implications of these results. Finally, I present the limitations of the current work and consider how future studies could address these weaknesses.

# 2. Material and methods

## 2.1. Data collection

In this study, I sought to gain a better understanding of the disparate effects of automation on the US job landscape using tools from statistics and network science. Data were drawn from three primary sources. Data on the automatability of individual jobs came from Frey & Osborne's 2017 study [12], in which they used a machine learning algorithm to classify automation likelihoods based on the tasks that comprised each job. Data on employment projections for each job came from the Bureau of Labor Statistics' 2016 report [19], which is publicly available at https://data.bls.gov/projections/occupationProj. Data on the importance of 120 skills, knowledge areas and abilities to each job were provided by O*Net. For each job, an importance score between 0 (low importance) and 100 (high importance) was assigned to every skill. Workers' ratings of the degree of current automation within jobs were also provided by O*Net.

Each of the three data sources labelled jobs using the Standard Occupational Classification (SOC) System. Data from distinct sources were therefore linked by matching the provided SOC codes. Jobs were included in the analysis if the information was available for (i) automatability, (ii) projected job growth, and (iii) skills, knowledge and abilities. This resulted in 683 jobs being included in the final sample, out of the 702 that were included in Frey & Osborne's study of automatability [12].

## 2.2. Network construction

For the creation of the network, nodes represented individual jobs and edges were given by the similarity of skill, knowledge and ability importance values between each pair of jobs. The similarity measure was

given by the following formula:

$$s_{ij} = \frac{1}{1 + d_{ij}}$$

and

$$d_{ij} = \sqrt{\sum (v_i - v_j)^2},$$

where $d_{ij}$ is the Euclidean distance between the vector of importance values for job $i$, $v_i$, and the vector of importance values for job $j$, $v_j$. Because this similarity metric is non-zero, it yields a fully connected network with 683 nodes and 232 903 edges. The validity of this similarity measure was tested by comparing pairwise job similarities with rates of worker transitions between 2011 and 2016, given by the Current Population Survey (CPS) [20]. To allow for direct comparisons, occupations with one-to-one correspondence between an SOC code (used to define network nodes) and an OCC code (used in the CPS data) were matched using the relevant crosswalk from the Bureau of Labor Statistics.

To obtain empirical skill-based job sectors, I performed community detection using an iterative generalized Louvain-like locally greedy algorithm [21]. This technique implements a stochastic optimization of the common modularity index value $Q$, in which nodes are individually reassigned to communities until no reassignment can improve $Q$. The modularity value, $Q$, of a network represents the degree of separation between nodes in different groups [22,23]. Intuitively, it measures the degree to which the network can be separated into non-overlapping communities with many, strong within-group connections and few, weak between-group connections. For a weighted network, in which edges have continuous numeric values, modularity can be defined as follows:

$$Q^w = \frac{1}{l^w} \sum_{i,j \in N} \left[ s_{ij} - \frac{S_i S_j}{l^w} \right] \delta_{m_i m_j},$$

where $S_i$ is the total strength of a node's edges, $l^w$ is the sum of all edge weights in the network, and $\delta_{m_i m_j}$ is 1 if $i = j$ and 0 otherwise. Importantly, it is well known that the modularity landscape suffers from a near degeneracy of optimal solutions [24]. Here, I addressed this issue by using 100 iterations of the Louvain-like algorithm. From the set of partitions obtained across these iterations, I built an agreement matrix from which I subsequently extracted a consensus partition [25].

## 2.3. Novel network measures

For this study, four new measures of job mobility were defined. The first is referred to as the *oasis coefficient*. This coefficient measures the extent to which a job is relatively safe from automation compared with jobs that require similar skills, knowledge and abilities. It was defined as,

$$O_i = \frac{\sum_{j \in N_i} a_j s_{ij}}{\sum_{j \in N_i} s_{ij}} - a_i,$$

where $N_i$ is the set of jobs most similar to node $i$ and $a_i$ is the automatability of job $i$. Thus, positive values indicate that a node is safer from automation than its neighbours, and negative values indicate that a node is more at-risk.

The second proposed metric is the *bridge coefficient*. This metric gives the extent to which a job's position in the network serves to connect highly automatable jobs to highly safe jobs, which are otherwise relatively divided. It was defined as,

$$B_i^l = IQR(A_{N_i}) \times (1 - p(A_{N_i})),$$

where $IQR(A_{N_i})$ is the interquartile range of the automatability values among jobs most similar to node $i$ and $p(A_{N_i})$ is a $p$-value giving the degree to which node $i$'s neighbouring jobs are divided by automatability. This measure is bounded between zero and one, with values close to one indicating that a node serves as a bridge between disparate regions of high and low automatability.

The third proposed metric is the *sector bridge coefficient*. Whereas the previous bridge coefficient quantified whether a job's position in the network connected low-automatability jobs to high-automatability jobs, this metric quantifies the extent to which a job has relatively strong skill-based connections to jobs in sectors other than its own. It was defined as,

$$B_i^s = \sum_{c \in C, c \neq c_i} \frac{\sum_{j \in N_c} s_{ij}}{n_c},$$

where $C$ is the set of all job communities or sectors, $c_i$ is the sector that contains job $i$, $N_c$ is the set of nodes in sector $c$, and $n_c$ is the number of nodes in sector $c$. Here, high values indicate strong connections outside of a job's own sector.

The final proposed metric is the *ladder coefficient*. This metric measures the extent to which a job's inter-sector connections are beneficial. In other words, it measures whether a job tends to be connected to the least automatable jobs in other sectors, or the most automatable jobs in other sectors. It was defined as,

$$L_i = -\rho(S_{ij,c_i \neq c_j} A_{j,c_i \neq c_j}),$$

where $\rho(x, y)$ is the Spearman correlation coefficient between $x$ and $y$, $S_{ij,c_i \neq c_j}$ is the set of edge weights between node $i$ and all nodes $j$ that are not within node $i$'s sector, and $A_{j,c_i \neq c_j}$ is the set of automatability values of all nodes $j$ that are not within node $i$'s sector. Since this coefficient is negated, high values represent stronger connections to safer jobs.

The first two proposed measures, the oasis and bridge coefficients, rely on specifying a job's local neighbourhood. For the primary analyses, this neighbourhood was taken to be the jobs in the 90th percentile or above in terms of their similarity to a given job. This resulted in neighbourhoods made up of 70 jobs, including the job for which the coefficients were being calculated. The effects of varying this neighbourhood size are shown in the electronic supplementary material, table S1.

To estimate the overall upward mobility of a job across the network, a mobility measure was operationalized as the sum of a job's own transition metrics and the degree to which the job is closely related to other jobs that are high on each transition metric. To obtain the degree to which a job's neighbours are high on a given metric, a weighted average is computed, with weights given by similarities to the central job. Then the measure of each job's overall upward mobility can be calculated using the following formula:

$$M_i = s(O_i^n) + s(B_i^l) + s(B_i^{l,n})$$
$$+ s(B_i^s) + s(B_i^{s,n}) + s(L_i) + s(L_i^n),$$

where $s(X_i)$ is the value of $X_i$ after being rescaled within a range of 0 to 1, and $X_i^n$ gives the weighted average of the relevant metric among neighbouring jobs. Notably, a job's own oasis coefficient is not included in the mobility measure, as this value does not represent a job's ability to transition to safe jobs and is necessarily highly related to the job's own automatability value.

## 2.4. Transition recommendation model

To understand the impact of the network on recommendations for retraining and job transition, I developed a model to estimate the expected benefit to workers that arises from following a set of transition recommendations. For a given job, the relative value of transitioning to other jobs can be framed as a combination of their safety from automation, the projected number of new jobs created between 2016 and 2026, and the degree to which they tend to have overlap in skills, knowledge areas and abilities with the job of interest. These measures give a rough estimate of the potential decrease in automatability, the scale of the transition opportunity and the ease of transitioning. From a given job $i$, the value of transitioning to job $j$, $t_j^i(w)$, is given by:

$$t_j^i(w) = w_a r(1 - a_j) + w_g r(e_j^g) + w_s r(s_{ij}),$$

where $r(x)$ is the rank of measure $x$ among all jobs other than job $i$, $e_j^g$ is the projected growth of job $j$ between 2016 and 2026, $s_{ij}$ is the similarity between job $i$ and job $j$ and $w$ is the set of $w_a$, $w_g$ and $w_s$, which give the relative weights of each component. To estimate the relative importance of each component when creating transition recommendations, the expected benefit of pursuing the transitions with the highest $t_j^i$ values can be optimized over the $w$.

The benefit of pursuing a transition from job $i$ to job $j$ was quantified as the decrease in automatability from job $i$ to job $j$, combined with binary markers that a worker (i) successfully completed retraining and (ii) was able to find and be hired in job $j$. This benefit is then,

$$b_{ij} = (a_i - a_j)\xi_{ij}\lambda_j,$$

$$\xi_{ij} \sim \text{Bern}(p_{ij}^r)$$

$$\text{and} \quad \lambda_j \sim \text{Bern}(p_j^h),$$

where $\xi_{ij}$ is a Bernoulli random variable representing the successful completion of retraining from job $i$ to job $j$, with probability $p_{ij}^r$, and $\lambda_j$ is a Bernoulli random variable representing successfully being hired for

job $j$, with probability $p_j^h$. In this study, the retraining probability was based on the similarity between job $i$ and job $j$, where higher similarity yielded a higher probability of successful retraining. The successful hiring probability was based on the projected growth of job $j$, where more open jobs yielded a higher probability of a successful hire. Given this benefit formula, the optimal weights of similarity, automatability and job growth were obtained by maximizing the expected overall benefit of someone in job $i$ making transition decisions based on transition scores that result from those weights. The full specification of this model can be found in the electronic supplementary material, methods.

## 2.5. Quantification of skill/automation associations

To determine which skills might be associated with higher or lower job automatability, Spearman correlations were calculated between a skill's importance across jobs and those jobs' automatability values. This measure was applied only to the 68 skills and knowledge areas, since these two categories offer more potential for purposeful retraining than abilities. Correlations were calculated across jobs within each sector, giving a measure of how associated individual skills and knowledge areas are with a sector's most or least automatable jobs. For a specific skill, $k$, and automatability measure, $a$, the correlation measure within sector $c$ is given by,

$$\rho_{rk,ra}^c = \frac{\text{cov}(rk_c, ra_c)}{\sigma_{rk_c}\sigma_{ra_c}},$$

where $rk_i$ is the rank of the importance of skill $k$ in job $i$, and $rk_c$ is the vector of ranks for all jobs within sector $c$. Though the analyses in the current study focus on examining skill–automation correlations within sectors, this measure can be easily extended to be job-specific by calculating the equivalent correlation within the neighbourhood of $n$ jobs most similar to the job of interest.

# 3. Results

In this study, I sought to gain a better understanding of the varying effects of automation across the US job landscape using tools from statistics and network science. Data came from three primary sources. First, I used data on the current and projected employment levels of each occupation in the US job market, provided by the Bureau of Labor Statistics. Second, I used the likelihood of automation for each job, based on research by Frey & Osborne [12]. Third, I used data on the importance of 120 skills, abilities and areas of knowledge to each job, provided by O*Net. To obtain a fully integrated job network, I calculated the similarity between each pair of jobs based on the set of 120 skills, abilities and areas of knowledge. For each job, an importance score between 0 (low importance) and 100 (high importance) was assigned to every skill. To calculate a similarity value in this space, I used a Euclidean distance-based similarity score (electronic supplementary material, methods). This measure was essentially the inverse of the Euclidean distance between the skill-wise importance values for a pair of jobs. Because every pair of jobs had some non-zero similarity, this technique yielded a fully connected network comprised of 683 nodes and 232 903 edges (figure 1).

The validity of this measure was tested by comparing the similarity of jobs $i$ and $j$ with the rate of transitions between job $i$ and job $j$ from 2011 to 2016, as reported by the Current Population Survey (CPS) [20]. Specifically, for each job, $i$, a correlation was calculated between its similarities to all other jobs and the rate at which workers transitioned between it and all other jobs (see the electronic supplementary material, methods, for more details). Jobs that did not have a one-to-one match between the SOC code used in O*Net data and the OCC code used in the CPS data were removed from this validation, as were jobs that had fewer than 100 total workers in the CPS sample. Among the 277 remaining jobs, the weighted average of their similarity-transition correlations (weighted by the number of survey participants who held each job at least once) was 0.43 (electronic supplementary material, figure S1). This suggests that the proposed similarity measure is indeed capturing the propensity of workers to transition between job pairs.

## 3.1. Finding job sectors

To understand the differential effect of automation on distinct sectors across the job landscape, I applied a community detection technique to the network [21]. This technique—a Louvain-like 'locally greedy' algorithm in which each cluster tries to claim nearby nodes as their own—partitions the network such that within-cluster connections are relatively strong and between-cluster connections are relatively

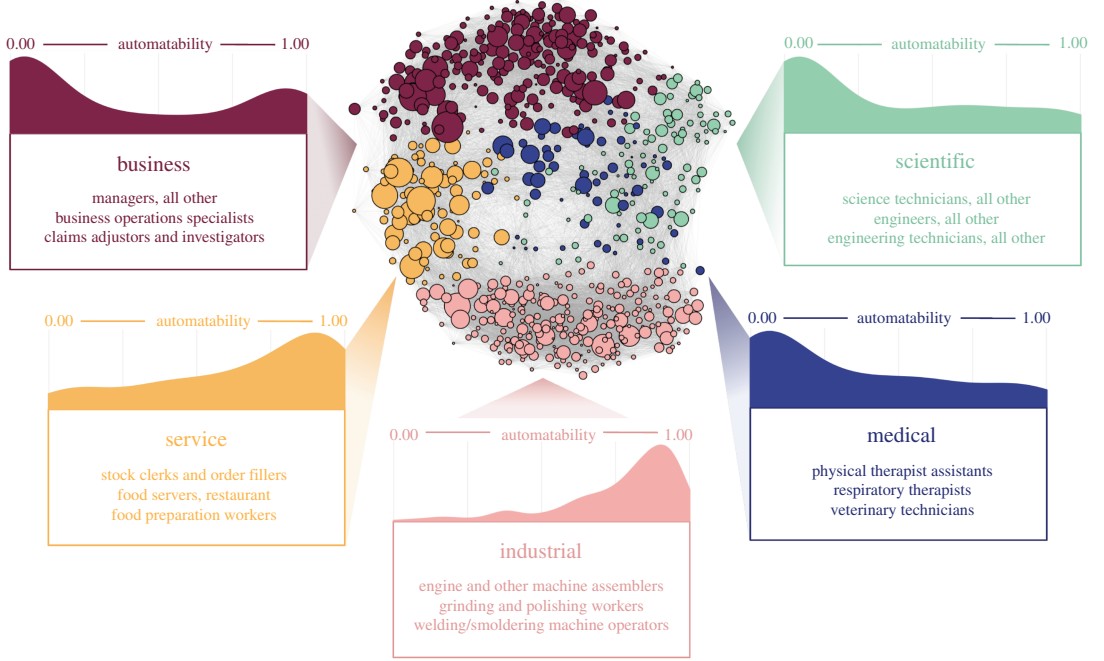

**Figure 1.** The US job network. Nodes ($N = 683$) represent individual jobs and edges ($E = 232\,903$) reflect the degree of skill, knowledge and ability overlap between each job pair. Nodes are coloured by sector affiliation and scaled by the number of workers in each job. Descriptions and characteristic jobs within each sector are placed along the graph, accompanied by the distribution of job automatability within each sector.

weak (see the electronic supplementary material, methods, for details). Due to the skill-based connections within the job network, the resulting clusters represent sectors in which jobs tend to be similar in terms of which skills, knowledge areas and abilities are important. From this process, five distinct sectors emerged (figure 1).

To understand what these sectors represent, characteristic jobs were found by calculating the degree to which each job's skill vector deviated from the average skill vector within its sector. The jobs with the least deviance from their sector's typical skill vector could then be described as the most characteristic jobs within that sector. The three most characteristic jobs from each sector can be found in figure 1. Based on these characteristic jobs, as well as further investigation into the skills, knowledge areas and abilities that are most unique to each sector, it was determined that the sectors broadly represented business-related, scientific, medical, service-related and industrial jobs.

Because these sectors are coarser than government-defined classifications, figure S2 shows the composition of CPS-defined occupation groups within each network-based sector. This comparison reveals that the business sector is largely made up of jobs from *office and administrative support*, *management, business and financial*, and *education, legal, community service, arts and media*; the scientific sector is largely made up of jobs from *computer, engineering and science*; the medical sector is largely made up of jobs from *healthcare practitioners and technical*; the service sector is largely made up of jobs from *service* and *production*; the industrial sector is largely made up of jobs from *production*, *construction and extraction*, and *installation, maintenance and repair*.

## 3.2. Impact of automation

Using the automation probabilities for each job, estimated by Frey & Osborne [12], I investigated the manner in which automation affects disparate regions of the job landscape (figure 1). Upon visual inspection, it appeared that the scientific and medical sectors tend to be populated by jobs with low levels of automatability, the business sector has a wider spread with some highly automatable and some highly safe jobs, and the service and industrial sectors tend to have mostly automatable jobs. The presence of between-sector differences was confirmed by an ANOVA, and Bonferroni-corrected (for all tests carried out in the study) pairwise $t$-tests revealed significantly higher automatability in the service and industrial sectors compared with the other three (all Bonferroni-corrected $p \ll 0.01$). No differences were found between the service and industrial sectors.

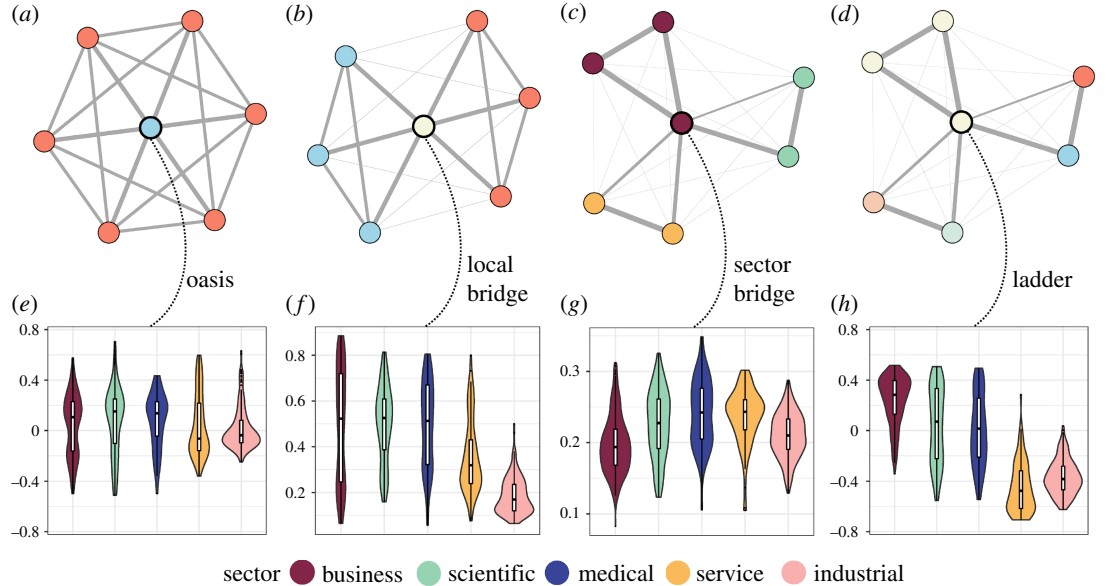

**Figure 2.** Measures of upward mobility within the job network. (*a*–*d*) Examples of oasis, local bridge, sector bridge and ladder jobs. (*a*) An oasis job (blue), which is significantly safer from automation than the jobs to which it is most similar. (*b*) A local bridge job (beige) provides a pathway from highly at-risk jobs to nearby safer jobs. (*c*) A sector bridge job (purple; centre) is strongly connected to jobs both within its own sector and within other sectors. (*d*) A ladder job (beige; centre) shows stronger inter-sector connections with safe jobs than automatable jobs. (*e*–*h*) Distributions of oasis, local bridge, sector bridge and ladder coefficients by sector.

These results appear to be consistent with the jobs on which the cultural discussion of automation seems to focus (e.g. truck drivers from the industrial sector, and cashiers from the service sector). Yet the automatability of an individual job represents only one piece of a worker's outlook. To build on this isolated effect, it is of interest to consider additional differences in workers' risk that arise from the degree to which they are or are not able to easily retrain for transitions to safer jobs. The framework of a connected job network provides an opportunity to do so through an examination of jobs' and sectors' patterns of connectivity.

## 3.3. Differential job mobility

### 3.3.1. Avoiding automation with local transitions

When considering how best to transition away from automation, it is potentially important to consider the necessity of switching sectors. Job sectors tend to be highly regional, meaning that cross-sector transitions may have an additional logistical burden, above and beyond the already difficult mental burden of learning the necessary new skills. As such, in this section I discuss two metrics for measuring a job's ability to serve as relief from automation for its most similar neighbouring jobs. The presence of these types of jobs within a sector would signal the potential for beneficial local mobility.

The first such metric is the *oasis coefficient*. As implied by its name, this coefficient measures the degree to which a job stands out within its neighbourhood as being uniquely safe from automation (figure 2*a*). There were no significant differences across sectors in their jobs' average oasis coefficients (figure 2*e*). This suggests that all sectors, regardless of automatability, have a similar distribution of jobs that are notably better or worse off than their neighbours. Yet importantly, in spite of there being no difference in the value of the oasis coefficients, oasis jobs within the industrial sector were found to have significantly higher automatability than oasis jobs within the business, scientific and medical sectors (all $p < 0.01$). No difference was found between the service sector and any of the other four sectors. This suggests that although the industrial sector has oasis jobs at similar rates to the other sectors, these oases still tend to be at higher risk of automation than comparable oases in other regions of the job landscape.

If industrial workers cannot rely on the prevalence of nearby automation oases, perhaps they can look for jobs that help ease them into other, safer regions of their sector. Along those lines, the second metric that yields insight into within-sector mobility is the *bridge coefficient*. This coefficient measures the degree

to which a job is closely related to both highly automatable and highly safe jobs, which are otherwise relatively divided (figure 2*b*). Here, significant differences were found between groups, with pairwise testing revealing that industrial and service jobs had significantly lower bridge coefficients than jobs in the remaining three sectors (figure 2*f*). Industrial jobs additionally had significantly lower bridge coefficients than service jobs (all $p \ll 0.01$). These results suggest that there may be fewer opportunities for local upwards mobility to safer regions of the job network within the service and industrial sectors, with this being particularly true within the industrial sector.

### 3.3.2. Avoiding automation with distant transitions

Based on the high automatability and the relatively few opportunities for local upwards mobility within the service and industrial sectors, the data seem to suggest that a reasonably large portion of displaced service and industrial workers may need to seek work in more dissimilar areas. To understand how this might be best accomplished, the inter-sector dynamics of the job network were investigated. As such, in this section, I discuss two metrics for measuring a job's ability to find relief from automation among relatively similar jobs outside of its own sector. The presence of these types of jobs within a sector would signal the potential for upwards mobility via movement across the job network.

The first metric is the *sector bridge coefficient*. Unlike the local bridge coefficient, which measured a job's ability to connect nearby low- and high-automatability jobs, the sector bridge coefficient measures the degree to which a job is well-connected to jobs in sectors other than its own (figure 2*c*). On this measure, some interesting patterns of connectivity appeared. Specifically, the business sector seemed to be the most insular, with its jobs showing significantly lower sector bridge coefficients than jobs in the scientific, medical and service sectors. The industrial sector was also somewhat insular, with significantly lower values than jobs in the medical and service sectors (figure 2*g*; all $p \ll 0.01$). These results suggest that workers in the scientific, medical and service sectors might have a slightly easier time translating their skills to jobs in other fields, compared with workers in the business and industrial sectors. This appears to be a positive result for workers in the service sector, where local measures of mobility were below average, and an unfortunate finding for workers in the industrial sector, indicating that their skills might not facilitate easy transitions to other sectors.

Yet the degree of connectivity or disconnectivity to other sectors in the job network only tells part of the story. One could imagine a job that has many strong inter-sector connections but all of its strongest connections are to highly automatable jobs, and another job that has few strong inter-sector connections but all of its strongest connections are to highly safe jobs. Along these lines, the second metric is the *ladder coefficient*. This metric measures the degree to which a node is most strongly connected to the safest jobs in other sectors (figure 2*d*). On this measure, clearer distinctions emerged in the feasibility of workers making inter-sector transitions to safe jobs. Notably, the business sector, while significantly lower than most other sectors on inter-sector connectivity, was significantly higher on the ladder coefficient than all other sectors (figure 2*h*). This suggests that jobs in business may not be very similar to jobs in other sectors, but the cross-sector jobs to which they are similar tend to be at very low risk of automation. Interestingly, although the service sector had relatively high connectivity to other sectors, both service and industrial jobs tended to be more strongly related to the most at-risk jobs in other sectors (all $p \ll 0.01$). This suggests that workers in these sectors may not benefit as much from making similarity-based transitions across fields.

### 3.3.3. The safe get safer

Compiling these four metrics for local and distant job transitions, a general measure of upward mobility was obtained for each job. Similar to the results found for the individual transition metrics, the service and industrial sectors had significantly lower upward mobility than the business, scientific and medical sectors. Additionally, the industrial sector showed significantly lower upward mobility than the service sector (all $p \ll 0.01$). Importantly, these differences all remained after accounting for jobs' automatability. Therefore, these results suggest that similarly at-risk workers in the service and industrial sectors may need more drastic shifts in their required skills, knowledge areas and abilities to transition to safe jobs compared with workers in other sectors. Interestingly, both before ($t_{681} = -20.91$, $p \ll 0.01$) and after ($t_{677} = -11.98$, $p \ll 0.01$) accounting for a job's sector, automatability was significantly negatively related to upward mobility (electronic supplementary material, figure S3). This result indicates that more automatable jobs tend to be significantly less able to move across the network into safe jobs. This relationship then demonstrates that the structure of the job network compounds the burden of automation, with highly automatable jobs also being the least capable of transitioning away from automation.

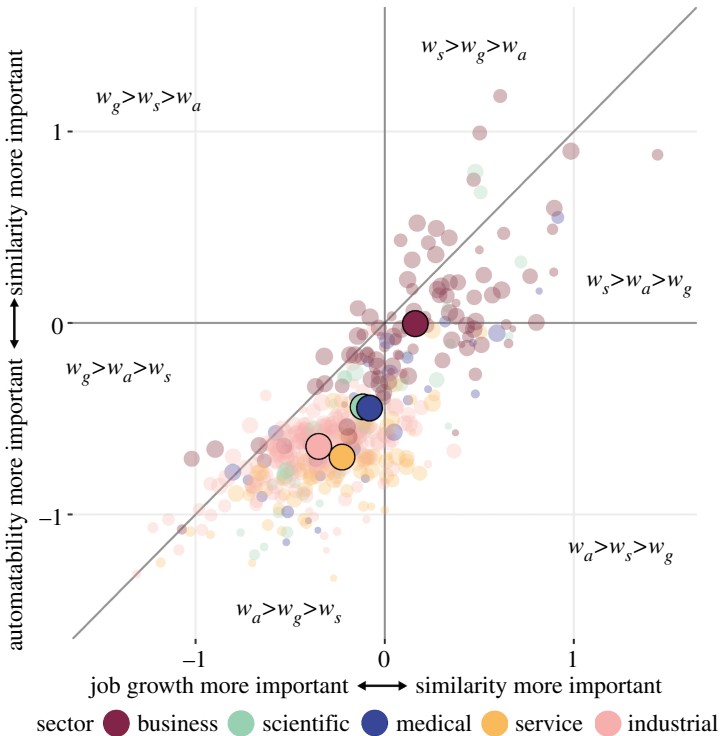

**Figure 3.** Relative importance of automatability, job growth and similarity for transition recommendations. The x-axis gives the log-ratio of $w_s$ to $w_g$, the y-axis gives the log-ratio of $w_s$ to $w_a$. Transparent points represent optimal relative weights for individual jobs, opaque points represent sector averages of optimal relative weights.

## 3.4. Optimal transitions for at-risk workers

If workers in the industrial and service sectors cannot reliably switch to safer jobs within their own fields and cannot easily transfer their skills to less automatable jobs in other sectors, then the question of how best to avoid automation remains. For that purpose, it is of interest to identify which jobs are worth the investment of extensive retraining for different workers and to investigate how these transition recommendations may vary across sectors. Three potential factors in deciding which transitions to pursue are (i) occupations' automatability, (ii) the number of available jobs, and (iii) jobs' similarity to a worker's current role.

Among these three factors, automatability and job growth are most commonly seen as important considerations for potential transitions, as evidenced by the frequency with which technology [26] and healthcare [27] jobs are recommended for a wide range of at-risk workers. Yet while incorporating similarity makes intuitive sense, the previous findings suggest that workers with less upward mobility may, in fact, be better off setting aside similarity in favour of moving farther across the job network to find safety. Therefore, it is critical to establish whether, and to what degree, similarity between two jobs should be considered when determining the potential benefit of a job transition. Additionally, it is important to understand whether the optimal formula for recommending job transitions varies based on a job's location within the network.

To understand the optimal weighting of different considerations when formulating transition recommendations, I optimized the expected benefit of a set of recommendations over $w_a$, the relative importance of automatability, $w_g$, the relative importance of growth and $w_s$, the relative importance of similarity, for each job. Because this optimization can break down for jobs with low automatability (which have few transitions capable of improving their automation prospects), results are presented for jobs with an automation probability greater than 0.25. Thus, these findings apply to the jobs from which workers will have at least a reasonable motivation to transition.

The relative importance of considering automatability, job growth and similarity in the transition recommendations for each job, given by the optimal values of $w_a$, $w_g$ and $w_s$, respectively, are shown in figure 3. Here, location along the x-axis is given by the log-ratio of $w_s$ and $w_g$ (where positive values indicate higher importance of similarity relative to growth), and location along the y-axis is given by the

**Table 1.** Best transition opportunities for heavy and tractor-trailer truck drivers. Rankings of the top 10 jobs based on safety from automation, projected job growth between 2016 and 2026, and skill, knowledge and ability similarity with truck drivers. Weighting of three components reflects the optimal 0.43 : 0.31 : 0.26 ratio found for heavy and tractor-trailer truck drivers.

| rank | occupation | automatability | projected growth | similarity |
|---|---|---|---|---|
| 1 | supervisors of mechanics, installers and repairers | 0.003 | 32 800 | 54.5 |
| 2 | electrical power-line installers and repairers | 0.097 | 16 800 | 93.9 |
| 3 | registered nurses | 0.009 | 438 100 | 34.6 |
| 4 | plumbers, pipefitters and steamfitters | 0.350 | 75 200 | 85.2 |
| 5 | recreation workers | 0.006 | 34 000 | 40.1 |
| 6 | emergency medical technicians and paramedics | 0.049 | 37 400 | 63.4 |
| 7 | electricians | 0.150 | 59 600 | 70.7 |
| 8 | preschool teachers | 0.007 | 50 100 | 34.0 |
| 9 | firefighters | 0.170 | 23 500 | 83.0 |
| 10 | police and sheriff's patrol officers | 0.098 | 47 800 | 63.5 |

log-ratio of $w_s$ and $w_a$ (where positive values indicate higher importance of similarity relative to automatability). Sector-specific average optimal values, weighted by each job's automatability, are given in table S2. These average optimal weights are also overlaid in figure 3.

Accounting for each job's automatability value, the optimal automatability weight for recommending transitions, $w_a$, was significantly lower for jobs in the business sector than those in all other sectors (all $p \ll 0.01$), and significantly higher for jobs in the service sector than those in all other sectors (all $p \ll 0.01$). The optimal job growth weight, $w_g$, was significantly higher for jobs in the industrial sector than those in the business and service sectors (all $p \ll 0.01$). The optimal similarity weight, $w_s$, was significantly higher for jobs in the business sector than those in all other sectors (all $p \ll 0.01$) and was significantly lower for jobs in the service and industrial sectors than those in the other three sectors (all $p \ll 0.01$). These results suggest that jobs with similar automation outlooks benefit from different types of transition recommendations, depending, in part, on where they are located within the job landscape. Specifically, even highly automatable jobs in the business sector tend to have enough mobility to prioritize easily attainable transitions to similar jobs, while automatable jobs in the service and industrial sectors may need to move farther across the landscape to find more dissimilar jobs with lower automatability and more expected growth. Yet interestingly, similarity was still a relatively important factor for jobs in all sectors.

These results suggest that while bigger jumps across the network may be necessary for jobs in more at-risk and less mobile sectors, relying on one-size-fits-all recommendations is still sub-optimal compared to making recommendations that take similarity into account. As an example, the top 10 recommendations for heavy and tractor-trailer truck drivers can be seen in table 1, reflecting the optimal weighting of $w_a = 0.43$, $w_g = 0.31$ and $w_s = 0.26$ that was found for this occupation. From this table, we see that the relatively low weight given to similarity results in high ratings for jobs with especially low automatability and high projected job growth, in spite of their lack of skill overlap with truck drivers. These jobs include some of the widely discussed recommendations in healthcare and teaching. Yet despite that relatively low similarity weight, many of the recommendations still represent transitions that have the potential to be more easily attainable. These include other jobs in the industrial sector, like power-line installers and repairers, plumbers and electricians, as well as managerial jobs like supervisors of mechanics.

## 3.5. Links between skills and automation

### 3.5.1. Skill development for job transitions

In practice, workers hoping to transition out of an at-risk job may be better off gaining a set of additional skills, or learning a new knowledge area, as opposed to retraining for one specific job. Thus, recommendations for specific job transitions may somewhat obscure the skills or knowledge that would be most beneficial to obtain. Here I sought to determine which skills might offer good opportunities for retraining by quantifying the association between each skill's importance across jobs

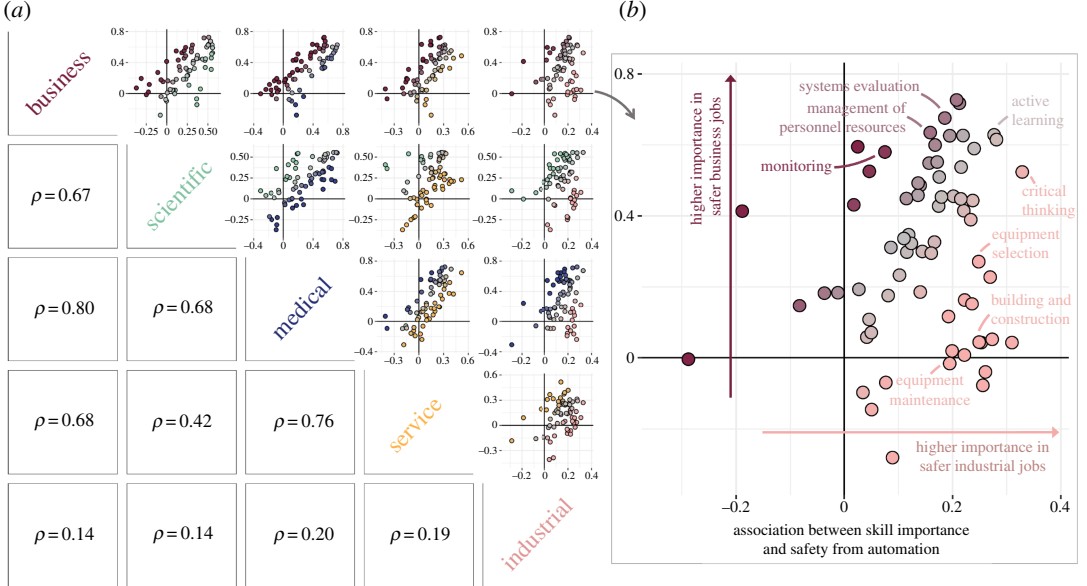

**Figure 4.** Relationships between skill importance and job automatability within and across sectors: (*a*) shows the degree to which sectors are similar with respect to the skills and knowledge areas most associated with safety from automation; (*b*) shows a detailed version of the scatterplot for the business and industrial sectors, highlighting skills that show a strong association with safety within one or both sectors.

and the jobs' degree of automatability. To find opportunities for training, I focus the following analyses solely on the 68 skills and knowledge areas, as they are likely to be more mutable than abilities.

To quantify the degree to which a specific skill is associated with safety from automation, I calculated the Spearman correlation between jobs' importance level for that skill and 1—jobs' automatability values. Positive values on this measure then reflect that less automatable jobs tended to place more importance on that skill, while negative values indicate that the skill tended to be more important in more automatable jobs. Because of the differences in skillsets across job sectors, I calculated these correlations separately within each sector.

Figure 4 shows how these associations vary across sectors. There is relatively high concordance in which skills are associated with lower automatability between the business, scientific, medical and service sectors, with pairwise Spearman correlations ranging from 0.42 to 0.80. Yet notably, the skills that are associated with lower automatability within the industrial sector tend to be different from the skills associated with lower automatability in the other four sectors, with correlations ranging from 0.14 to 0.20. Figure 4 also shows a detailed plot of skills' benefit within the industrial sector compared with their benefit within the business sector. This figure shows that some soft skills, like *active learning* and *critical thinking*, tend to be associated with lower automatability in both sectors, while skills like *monitoring* and *management of personnel resources* tend to be more beneficial within the business sector, and skills like *equipment maintenance* and *building and construction* tend to be more beneficial within the industrial sector. Table 2 shows the five skills most associated with lower automatability within each sector.

### 3.5.2. Skill redefinition within jobs

Facilitating a transition to a new job may be important in some cases, but the automation of certain tasks within an occupation may not always indicate that a transition is necessary. Indeed, some jobs facing future automation have been shown to incorporate less automatable skills and tasks into the same job without decreasing employment. Though the cross-sectional nature of these data make it difficult to fully capture the process of occupational skill redefinition, workers' assessments of the degree of existing automation within a job, as provided by O*Net, can give insight into jobs for which the process of task automation has already begun.

Unlike future automatability, for which jobs in the service and industrial sectors showed significantly higher values than those in the other three sectors, current automation levels were similar across sectors. Aside from jobs in the service sector having a slightly lower mean automation value than jobs in the business sector ($p < 0.05$), no other pairwise comparisons were significant. Additionally, the overall

**Figure 5.** Relationship between degree of current automation and susceptibility to future automation across jobs: (*a*) shows the relationship for all jobs in the data; (*b*–*f*) show the relationships within each sector and highlights notable jobs. Solid lines demarcate the average values across all jobs for current automation and future automatability.

**Table 2.** Skills most associated with safety from automation. Rankings of the top five skills within each sector, with respect to their estimated correlations with 1—automatability.

| rank | business | scientific | medical |
|------|----------|------------|---------|
| 1 | learning strategies | judgement and decision-making | active learning |
| 2 | instructing | critical thinking | judgement and decision-making |
| 3 | systems evaluation | active learning | psychology |
| 4 | management of personnel resources | persuasion | critical thinking |
| 5 | active learning | operations analysis | social perceptiveness |

| rank | service | industrial |
|------|---------|------------|
| 1 | therapy and counselling | critical thinking |
| 2 | biology | law and government |
| 3 | psychology | complex problem-solving |
| 4 | medicine and dentistry | active learning |
| 5 | philosophy and theology | transportation |

Spearman correlation between current automation and future automatability across all jobs was only 0.28, suggesting that the jobs that currently involve the most automation are not necessarily the ones most likely to face future automation (figure 5).

To obtain preliminary estimates of which skills might be candidates for occupational skill redefinition in the face of future automation, I sought to find skills that were both negatively associated with future automatability and positively associated with current automation. Such skills, therefore, represent those

**Table 3.** Skills most associated with both current automation and safety from future automation. Rankings of the top five skills within each sector, with respect to the average of their estimated correlations with current automation and 1—future automatability.

| rank | business | scientific | medical |
|------|----------|------------|---------|
| 1 | systems evaluation | systems analysis | reading comprehension |
| 2 | systems analysis | management of personnel resources | instructing |
| 3 | judgement and decision-making | judgement and decision-making | critical thinking |
| 4 | monitoring | management of financial resources | programming |
| 5 | complex problem-solving | systems evaluation | active listening |

| rank | service | industrial |
|------|---------|------------|
| 1 | law and government | monitoring |
| 2 | clerical | operation monitoring |
| 3 | equipment maintenance | biology |
| 4 | economics and accounting | operation and control |
| 5 | public safety and security | science |

that are important within jobs that have already undergone automation and are estimated to be relatively safe from future automation. Associations between skill importance and each automation measure were calculated similarly to those described in the previous section, but the relationship between jobs' current automation and their future automatability was controlled by using partial Spearman correlations.

Table 3 shows the five skills most highly associated with both current automation and future safety within each sector (see figure S7 for a visualization of the relationships between these two components). While there is some overlap with the skills that are most associated with future automatability, considering skills' relationships with current automation seems to highlight those that can be directly incorporated into a newly automated job. For example, when only looking at safety from future automatability, three of the top five skills within the industrial sector were soft skills like *critical thinking* and *active learning*. Using this composite measure, however, three of the top five appear to be more technical skills related to oversight of machinery (e.g. *operation monitoring*).

# 4. Discussion

There is a vibrant ongoing discussion being carried out by academics, think tanks and journalists regarding the potential effects of job automation and the ways in which its harm may be mitigated. Yet within this discussion, individual occupations are often treated as independent units, as opposed to pieces of an integrated system that workers can move across. This framing impacts the way people consider the potential harm of automation—highlighting the automatability of each job without considering the ability of its workers to find new opportunities—and the way people recommend and facilitate job transitions for at-risk workers—directing at-risk workers to a handful of the most promising jobs without considering more tailored solutions. To address these gaps in the automation discussion, the current study conceptualizes the job market as a network of occupations with varying degrees of similarity in the skills, knowledge and abilities they require. Using this framework, the presence of differential upward mobility among automatable jobs is investigated, a method for finding optimal tailored transitions for each job is presented, and the potential benefits of individual skills are estimated.

## 4.1. Differential job mobility

The automation outlook of a given job is often thought of as being exclusively a function of that particular job's likelihood of being automated. Yet this isolated view may not fully represent the ability of workers in that occupation to find a successful outcome in the face of automation. By considering the mobility of workers across the job market and into less automatable jobs, this study sought to gain a more complete understanding of how the automation of individual jobs may affect workers.

Using a composite measure of jobs' upward mobility, this study revealed two important findings. First, across all sectors, more automatable jobs tend to have less upward mobility. This suggests that the structure of the job network tends to make the outlook of automatable jobs poorer while making the outlook of safe jobs even better. Yet the second finding revealed that, after accounting for automatability, jobs in the business, scientific and medical sectors were more mobile than those in the service sector, which were more mobile than those in the industrial sector. This then implies that workers in jobs with a 90% chance of automation within the business, scientific and medical sectors will likely have an easier time finding safe employment opportunities than workers in jobs with an equal 90% chance of automation within the service and industrial sectors. Overall, these findings begin to answer the question of which at-risk jobs may need more or less help finding and completing job transitions.

Although these sector-level discrepancies are in line with the common thinking that blue-collar jobs are most at risk [9,28], recently the perception has been spreading that white-collar jobs may be equally affected [29,30]. Because these views have typically been based solely on the automatability of individual jobs, the findings presented in this study offer a new layer for considering a job's outlook. With this framework, it now seems that the links between jobs make white-collar workers—even those with highly automatable occupations—at least somewhat safer than their blue-collar counterparts. If governments, non-profits and businesses are going to be appropriately mobilized to aid workers at the greatest risk of long-term displacement, it is therefore important to consider mobility alongside automatability.

## 4.2. Optimal transitions for at-risk workers

Importantly, understanding which jobs will have a harder time successfully completing job transitions is only half of the story. Once at-risk jobs are identified, it is crucial to determine which transitions will be the most beneficial for workers seeking to make a switch. At this moment in time, the common wisdom regarding job transitions appears to be that workers in automatable occupations should consider retraining for jobs in technology [26] or healthcare [27]. As these fields are safe from automation [12] and are growing rapidly [19], these recommendations make ostensible sense. Yet there are many barriers facing workers who want to switch jobs [14–16], and it is possible that technology and healthcare would represent some of the most difficult transitions possible for many at-risk workers. Therefore, this study sought to determine whether workers benefit from tailored recommendations that consider not only automatability and growth but also the similarity of skillsets.

To accomplish this, I proposed a model to estimate the expected automatability decrease for workers in a given job upon attempting a set of recommended transitions. By varying the recommendations based on the relative importance of automatability, growth and similarity, this model revealed that considering similarity when presenting transition recommendations improved expected outcomes at least somewhat for all jobs. Importantly, the relative importance of automatability, growth and similarity varied based on sector, with similarity being slightly less beneficial for service and industrial jobs. This difference is likely a reflection of the lower mobility of these jobs, which results in them needing to move farther across the network to find good options. Yet broadly, these results point to the importance of resisting one-size-fits-all recommendations. Instead, it appears beneficial to consider which jobs present safe and growing opportunities while also using a worker's existing skills and minimizing the difficulty of retraining.

Indeed, much of the 'future of work' discussion occurring within non-profits and think tanks seems to align with these findings. The rapidly growing share of older workers in the workforce [31] points to the potential importance of transitions that do not require technological ability or additional degrees, and the high value workers place on job stability [31] may be a reason for them to remain in fields where they are already somewhat established. Additionally, while 72% of workers believe they have a personal responsibility to acquire job skills [32], programmes to facilitate this skill-building have not seen improved employment outcomes among their participants [14,33]. A review of one such model theorized that this may be due to a lack of customized advice, and an inability to learn and incorporate applicants' prior skills and qualifications [33]. Although a worker's current job is a crude measure of existing skills, the proposed method for obtaining similarity-based recommendations could be a step towards straightforward customization of retraining advice and could easily be applied to more accurate skill measures like self-assessment [34].

## 4.3. Links between skills and automation

Even when tailored recommendations of job transitions can be obtained, workers may be better off increasing a set of skills that are resistant to automation rather than retraining for specific jobs. To determine which skills could be beneficial in this context, I assessed the association between skills'

importance across jobs and the automatability of those jobs. There was notable variability across sectors in the skills that were most associated with safety from automation, suggesting that jobs' location within the network may affect which skills are most valuable. Industrial jobs showed the largest cross-sector differences, such that the skills most associated with non-automatable industrial jobs had little overlap with the skills most associated with non-automatable jobs in other sectors. Yet in general, skills that were associated with lower automatability tended to be cognitive and social skills (e.g. *critical thinking*, *active learning*, *social perceptiveness*). This is consistent with previous findings demonstrating the increasing value of soft skills in the labour market [35].

However, some work has shown that in the face of computerization, many jobs do not reduce employment (thus requiring job transitions) but instead redefine the tasks of the job [36,37]. Indeed, by looking at the degree to which jobs were already automated, I found that current automation showed a relatively low association with future automatability. This suggests that some jobs that are already highly computerized have incorporated tasks and skills that are not likely to be automatable going forward. While skills that tend to be associated with less automatable jobs may be candidates for within-job skill redefinition, they may also be specific to jobs that will not face computerization (e.g. *fine arts* is associated with low automatability, yet it is unlikely to be commonly incorporated into highly computerized jobs).

To look closer at the skills that may see increased value within automatable jobs as they become more computerized, I assessed skills relationships to both future automatability and current automation levels. While these measures do not capture longitudinal changes within jobs, skills that are associated with low future automatability and high current automation may reflect those that become more important as highly computerized jobs adapt. Interestingly, skills that fit this pattern of association tended to be a bit more specific to the individual sectors and were occasionally more technical than the skills found by looking at future automatability alone. However, the skills that ranked highly on this measure still tended to be associated with problem-solving (e.g. *systems evaluation*, *complex problem-solving*) and communication (e.g. *instructing*, *management of personnel resources*). This is consistent with findings from case studies and longitudinal research that investigated skill changes following the adoption of new technologies [38,39].

## 4.4. Limitations

The methods and conclusions presented in this work are subject to several limitations. First, the analyses presented here are quite dependent on the estimates of automatability, the projections of job growth and the importance values given to job skills that were obtained by external researchers and organizations. As these research areas develop and as the values change based on the changing job market, new estimates can be easily incorporated into the framework presented here. With respect to automatability, various studies have proposed estimating jobs' susceptibility to automation at an individual level rather than an occupational level [5,40,41]. As these studies have generally resulted in lower estimates of automatability than those found by Frey & Osborne [12], it would be quite valuable for future work to develop methods for incorporating individual- and job-level heterogeneity into the proposed occupation network framework.

Second, because of the preliminary nature of the job transition model, the specification of how retraining success related to job similarity and employment success related to job growth was somewhat arbitrary. Because of the lack of existing research on the probability of successfully transitioning from any job to any other job, the effects of varying these specifications across a range of possibilities are presented in the electronic supplementary material, figures S4–S6. Yet it is equally important to note that as future research engages with these questions, it will be straightforward to incorporate updated values into the model to obtain better recommendations.

Finally, despite the relatively strong association between pairwise job similarities and observed job transitions, there are various considerations beyond skill overlap that would affect the practical similarity between jobs. Future work could develop ways of building on the proposed similarity measure to incorporate the additional factors that may impact transition ability, like a discrepancy between required degrees or a lack of regional overlap.

## 4.5. Conclusion

As the potential for large-scale automation increases, it becomes more important to understand the risk workers face and the ways in which they can best avoid unexpected displacement. Although many public

and private organizations are doing great work to provide workers with relevant knowledge, the recommendations brought up in the public discussion are often overly broad, focused on flashy new careers, and ignorant of easier job transitions that use more of workers' existing expertise. The current study sought to investigate the integrated nature of the job market in order to gain a better understanding of how automation may differentially affect workers in different sectors and how transition and reskilling recommendations can best be tailored to workers' unique situations.

Overall, I found that at-risk workers in the business, scientific and medical sectors are generally well-equipped to avoid automation by taking advantage of accessible job transitions. Yet the data suggest that existing skills among workers in the service and industrial sectors provide less opportunity for transitions to safer jobs. As such, I presented a model for understanding how workers may benefit from different types of transition recommendations and determined that optimal recommendations should not only consider a new job's automatability and growth potential, but also its similarity to the job a worker is leaving. Additionally, I found that the skills most associated with less automatable jobs differ across sectors, and skills that facilitate job transitions may be distinct from those that facilitate within-job skill redefinition. Overall, this study demonstrates that automation will likely have complex effects on the job market that are not fully captured by the likelihood of individual jobs becoming automated. Instead, it is important that policy-makers, businesses and workers consider the relationships between jobs when determining who is at the highest risk of long-term displacement and which transition or reskilling opportunities they should pursue.

Data accessibility. Data and code for this work are stored in the Open Science Framework repository and can be accessed at https://goo.gl/2xbmQH [42].

Competing interests. The author declares that he has no competing interests.

Funding. The author has no funding to declare.

Acknowledgements. The author would like to thank Aurora Jensen for vital contributions to the formulation of this project, Ilia Blinderman for assistance in reviewing the literature and conceptualizing the issues discussed in this piece, Danielle S. Bassett and Russell T. Shinohara for valuable feedback on the manuscript, and the two anonymous reviewers whose comments helped improve and clarify this work.

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
