## [Reviewer comments · Royal Society Open Science]

Review History

RSOS-182124.R0 (Original submission)

Review form: Reviewer 1

Is the manuscript scientifically sound in its present form?

Yes

Are the interpretations and conclusions justified by the results?

No

Is the language acceptable?

Yes

Is it clear how to access all supporting data?

Yes

Do you have any ethical concerns with this paper?

No

Have you any concerns about statistical analyses in this paper?

No

Recommendation?

Major revision is needed (please make suggestions in comments)

Comments to the Author(s)

This article maps the space of US job titles to identify which sectors contain workers who are (1) susceptible to automation and (2) stuck without a pathway to alternative employment. For example, the author finds that occupations in both the Business and Industrial sectors may be susceptible to automation but the Industrial sector on the whole has less connectivity to other sectors. This type of result may inform our understanding of technological change and job polarization in the US. I love this idea, and I believe there is great value in performing this analysis well.

I have a few major concerns with this article.

(1) The author's interpretation of the edge weights in the job network are not validated strongly enough for the interpretation the author assigns to them. The author proposes a single measure for the skill similarity between a pair of jobs and considers this similarity as a proxy for workers' ability to transition between those occupations without showing that these scores are even correlated with worker career mobility. Despite what the author says in the Limitations section, data does exist to approximate this. One possibility is to access resume data. Another possibility is to use data from the Current Population Survey, which is a data product of the US Census Bureau. Comparing the edge weights with actual rates of workers transitioning between occupation pairs would strengthen the author's interpretation of the job network.

(2) Similar to (1), the author should validate their measures for connectivity between sectors against empirical data reflective of actual worker transitions. The author's use of job growth per sector offers an indirect validation, and I believe a more direct validation is possible.

(3) This study relies entirely on the automation estimates from the Frey & Osborne study, but this study is highly controversial. Therefore, the author should strengthen their claims by showing how their results change when different sets of estimates are considered. For this purpose, I recommend also considering

- O*NET Degree of Automation scores, and
- automation estimates from the Arntz et al study (ref. [5] in the article).

(4) Although workers care about their jobs and policy makers care about employment numbers, work from Autor, Acemoglu, and others all suggest that occupations are best understood as abstract bundles of tasks and skills because technology interacts with labor by impacting the demand for specific tasks and skills. Therefore, does an occupation's susceptibility to automation impact the occupation's skill similarity to other occupations? In turn, does this fundamentally change the job network?

Work from James Bessen (BU) and, alternatively, from David Deming (Harvard) demonstrate that occupations that are exposed to automation can redefine their skill requirements to reflect changing labor demands. Often, these transitions focus on an increased reliance on social skills. Therefore, exposure to automating technology does not always reduce employment. As an example from Bessen's work, the increased adoption of ATMs corresponded to national increase in employment for bank tellers in the US--perhaps a counter-intuitive result--because ATMs made it cheaper to open bank branches (i.e., demand elasticity) and bank tellers began to replace clerical skills with social skills. So, since this occupational skill redefinition has been shown to

occur with exposure to automating technology, how does the author incorporate these insights into our understanding of the job network in the study?

(5) If the skill similarity scores that define the edge weights in the job network do indeed correspond to worker transitions between occupations, then which skills promote career mobility for workers? Are these the skills that are susceptible to automation (this may be difficult to answer convincingly)? More practically, this type of insight would be helpful to policy makers attempting to promote career mobility for workers through informed worker retraining programs. This level of insight should significantly strengthen the impact of this study.

Minor comments:

- the "locally greedy" clustering algorithm employed in this study is often referred to as "Louvain community detection" in my experience
- Figure 2 is very helpful for clarifying the measures.
- on page 8, it is not clear to me what variable t is referring to. Is it the same t that is defined on page 9?
- Table 2 represents an excellent level of insight (of course, I have my concerns listed above). This is a level of detail that policy makers should find very helpful.

Review form: Reviewer 2 (Olga Scrivner)

Is the manuscript scientifically sound in its present form?

Yes

Are the interpretations and conclusions justified by the results?

Yes

Is the language acceptable?

Yes

Is it clear how to access all supporting data?

Yes

Do you have any ethical concerns with this paper?

No

Have you any concerns about statistical analyses in this paper?

No

Recommendation?

Accept with minor revision (please list in comments)

Comments to the Author(s)

The study investigates the effects of automation on labor force and creates a network of jobs by their skills similarities. The identification of automation is based on the classification and the network results show distinct patterns of automation by different job sectors. In addition, the study introduces metrics for measuring job mobility. Finally, it offers a solution formula to estimate the cost/benefit of transitioning. Overall, the novel techniques and findings contribute to the field and provide novel solutions to workforce decision-makers. However, the paper suffers from a weaknesses in the methodology and analysis sections, mostly due to the lack of clarity and more detailed information.

Suggestions

- Data section and methodology should be presented separately and prior to the analysis section.
- Please detail the size of BLS data, the number of occupations, the type of occupational code (e.g. SOC), how data was extracted (e.g. link to the BLS data) or mention that it is provided in supplemental materials.
- It is not clear how BLS data, Frey's data, and ONET data are merged. Cite O*Net, BLS. Consider using the real job postings data with skills (see EMSI, BGT etc) in the future.

Figure 1.

- How the number of clusters was determined? Medical cluster does not seem to be well-defined.
- It is not clear how sectors were assigned, for example using NAICS cross-walk?
- Note also that in Frey's study, business was defined as low risk for automation, in Figure 1 business shows a bimodal distribution - perhaps two different categories?

Differential job mobility

- The reviewer suggests to introduce briefly all 4 measures at the beginning of the section. Consider also adding labels to each figure, for example "local" or within and "distant" or inter
- For Figure 2 E-H - Consider adding y-axis labels and median
- If positive values in 2F indicate a bridge, then all 4 sectors are bridges [all positive]?
- 2G - not clear how business and industrial have lower bridge coefficients - is it based on their median?

Could the bi-modality of business affect the inter-sector connectivity?

page 2 line 32 - spelling
page 13 line 50 - spelling

Decision letter (RSOS-182124.R0)

04-Apr-2019

Dear Mr Dworkin,

The editors assigned to your paper ("Network-driven differences in mobility and optimal transitions among automatable jobs") have now received comments from reviewers. We would like you to revise your paper in accordance with the referee and Associate Editor suggestions which can be found below (not including confidential reports to the Editor). Please note this decision does not guarantee eventual acceptance.

Please submit a copy of your revised paper before 27-Apr-2019. Please note that the revision

deadline will expire at 00.00am on this date. If we do not hear from you within this time then it will be assumed that the paper has been withdrawn. In exceptional circumstances, extensions may be possible if agreed with the Editorial Office in advance. We do not allow multiple rounds of revision so we urge you to make every effort to fully address all of the comments at this stage. If deemed necessary by the Editors, your manuscript will be sent back to one or more of the original reviewers for assessment. If the original reviewers are not available, we may invite new reviewers.

- Data accessibility

<http://datadryad.org/submit?journalID=RSOS&manu=RSOS-182124>

- Competing interests

- Authors' contributions

All submissions, other than those with a single author, must include an Authors' Contributions section which individually lists the specific contribution of each author. The list of Authors should meet all of the following criteria; 1) substantial contributions to conception and design, or

acquisition of data, or analysis and interpretation of data; 2) drafting the article or revising it critically for important intellectual content; and 3) final approval of the version to be published.

- Acknowledgements

- Funding statement

on behalf of Dr Robert MacKay (Associate Editor) and Mark Chaplain (Subject Editor)
openscience@royalsociety.org

Associate Editor's comments (Dr Robert MacKay):

The reviewers found the paper interesting but both made substantial suggestions for improvement. We invite you to submit a revised version to take their comments into account.

Comments to Author:

Reviewers' Comments to Author:

Reviewer: 1

Comments to the Author(s)

This article maps the space of US job titles to identify which sectors contain workers who are (1) susceptible to automation and (2) stuck without a pathway to alternative employment. For example, the author finds that occupations in both the Business and Industrial sectors may be susceptible to automation but the Industrial sector on the whole has less connectivity to other sectors. This type of result may inform our understanding of technological change and job polarization in the US. I love this idea, and I believe there is great value in performing this analysis well.

I have a few major concerns with this article.

(1) The author's interpretation of the edge weights in the job network are not validated strongly enough for the interpretation the author assigns to them. The author proposes a single measure for the skill similarity between a pair of jobs and considers this similarity as a proxy for workers' ability to transition between those occupations without showing that these scores are even correlated with worker career mobility. Despite what the author says in the Limitations section, data does exist to approximate this. One possibility is to access resume data. Another possibility is to use data from the Current Population Survey, which is a data product of the US Census Bureau. Comparing the edge weights with actual rates of workers transitioning between occupation pairs would strengthen the author's interpretation of the job network.

(2) Similar to (1), the author should validate their measures for connectivity between sectors against empirical data reflective of actual worker transitions. The author's use of job growth per sector offers an indirect validation, and I believe a more direct validation is possible.

(3) This study relies entirely on the automation estimates from the Frey & Osborne study, but this study is highly controversial. Therefore, the author should strengthen their claims by showing how their results change when different sets of estimates are considered. For this purpose, I recommend also considering

- O*NET Degree of Automation scores, and
- automation estimates from the Arntz et al study (ref. [5] in the article).

(4) Although workers care about their jobs and policy makers care about employment numbers, work from Autor, Acemoglu, and others all suggest that occupations are best understood as abstract bundles of tasks and skills because technology interacts with labor by impacting the demand for specific tasks and skills. Therefore, does an occupation's susceptibility to automation impact the occupation's skill similarity to other occupations? In turn, does this fundamentally change the job network?

Work from James Bessen (BU) and, alternatively, from David Deming (Harvard) demonstrate that occupations that are exposed to automation can redefine their skill requirements to reflect changing labor demands. Often, these transitions focus on an increased reliance on social skills. Therefore, exposure to automating technology does not always reduce employment. As an example from Bessen's work, the increased adoption of ATMs corresponded to national increase in employment for bank tellers in the US--perhaps a counter-intuitive result--because ATMs made it cheaper to open bank branches (i.e., demand elasticity) and bank tellers began to replace clerical skills with social skills. So, since this occupational skill redefinition has been shown to occur with exposure to automating technology, how does the author incorporate these insights into our understanding of the job network in the study?

(5) If the skill similarity scores that define the edge weights in the job network do indeed correspond to worker transitions between occupations, then which skills promote career mobility for workers? Are these the skills that are susceptible to automation (this may be difficult to answer convincingly)? More practically, this type of insight would be helpful to policy makers attempting to promote career mobility for workers through informed worker retraining programs. This level of insight should significantly strengthen the impact of this study.

Minor comments:

- the "locally greedy" clustering algorithm employed in this study is often referred to as "Louvain community detection" in my experience
- Figure 2 is very helpful for clarifying the measures.

- on page 8, it is not clear to me what variable t is referring to. Is it the same t that is defined on page 9?
- Table 2 represents an excellent level of insight (of course, I have my concerns listed above). This is a level of detail that policy makers should find very helpful.

Reviewer: 2

Comments to the Author(s)

The study investigates the effects of automation on labor force and creates a network of jobs by their skills similarities. The identification of automation is based on the classification and the network results show distinct patterns of automation by different job sectors. In addition, the study introduces metrics for measuring job mobility. Finally, it offers a solution formula to estimate the cost/benefit of transitioning. Overall, the novel techniques and findings contribute to the field and provide novel solutions to workforce decision-makers. However, the paper suffers from a weaknesses in the methodology and analysis sections, mostly due to the lack of clarity and more detailed information.

Suggestions

- Data section and methodology should be presented separately and prior to the analysis section.
- Please detail the size of BLS data, the number of occupations, the type of occupational code (e.g. SOC), how data was extracted (e.g. link to the BLS data) or mention that it is provided in supplemental materials.
- It is not clear how BLS data, Frey's data, and ONET data are merged. Cite O*Net, BLS. Consider using the real job postings data with skills (see EMSI, BGT etc) in the future.

Figure 1.

- How the number of clusters was determined? Medical cluster does not seem to be well-defined.
- It is not clear how sectors were assigned, for example using NAICS cross-walk?
- Note also that in Frey's study, business was defined as low risk for automation, in Figure 1 business shows a bimodal distribution - perhaps two different categories?

Differential job mobility

- The reviewer suggests to introduce briefly all 4 measures at the beginning of the section. Consider also adding labels to each figure, for example "local" or within and "distant" or inter
- For Figure 2 E-H - Consider adding y-axis labels and median
- If positive values in 2F indicate a bridge, then all 4 sectors are bridges [all positive]?
- 2G - not clear how business and industrial have lower bridge coefficients - is it based on their median?

Could the bi-modality of business affect the inter-sector connectivity?

page 2 line 32 - spelling

page 13 line 50 - spelling

Author's Response to Decision Letter for (RSOS-182124.R0)

See Appendix A.

Decision letter (RSOS-182124.R1)

23-May-2019

Dear Mr Dworkin,

I am pleased to inform you that your manuscript entitled "Network-driven differences in mobility and optimal transitions among automatable jobs" is now accepted for publication in Royal Society Open Science.

on behalf of Dr Robert MacKay (Associate Editor) and Mark Chaplain (Subject Editor)
openscience@royalsociety.org

Associate Editor Comments to Author (Dr Robert MacKay):
Associate Editor

Comments to the Author:

The author has taken detailed account of the reviewers' comments and I recommend the paper for publication now.

Follow Royal Society Publishing on Twitter: [@RSocPublishing](https://twitter.com/RSocPublishing)

Appendix A

Response to reviewers for “Network-driven differences in mobility and optimal transitions among automatable jobs”

Jordan D. Dworkin^a

^aDepartment of Biostatistics, Epidemiology, & Informatics, Perelman School of Medicine, University of Pennsylvania, Philadelphia, PA, USA

To the Editor:

I would like to thank the editor and the reviewers for your consideration and evaluation of the manuscript “Network-driven differences in mobility and optimal transitions among automatable jobs.” I very much enjoyed engaging with the reviewers’ thoughtful and thorough comments, and I believe that the manuscript has been greatly improved.

In this document, I give my responses to the reviewers’ comments point-for-point, and describe the changes made within the manuscript. Numbered comments from the reviewers are presented, and my responses to those comments follow (within dashed lines for clarity).

I appreciate the opportunity to incorporate these suggestions into the manuscript, and I am pleased to resubmit it for further consideration by *Royal Society Open Science*. Please do not hesitate to contact me if I can provide further information or clarification.

Yours Sincerely,

Jordan Dworkin
University of Pennsylvania

Reviewer 1

Reviewer 1 summary:

This article maps the space of US job titles to identify which sectors contain workers who are (1) susceptible to automation and (2) stuck without a pathway to alternative employment. For example, the author finds that occupations in both the Business and Industrial sectors may be susceptible to automation but the Industrial sector on the whole has less connectivity to other sectors. This type of result may inform our understanding of technological change and job polarization in the US. I love this idea, and I believe there is great value in performing this analysis well.

I have a few major concerns with this article.

I would like to thank the reviewer for her/his incredibly thoughtful and thorough comments. I believe that these suggestions have greatly improved the manuscript, and appreciate the reviewer's time in carefully considering the work.

Reviewer 1 comments:

Comment 1: The author's interpretation of the edge weights in the job network are not validated strongly enough for the interpretation the author assigns to them. The author proposes a single measure for the skill similarity between a pair of jobs and considers this similarity as a proxy for workers' ability to transition between those occupations without showing that these scores are even correlated with worker career mobility. Despite what the author says in the Limitations section, data does exist to approximate this. One possibility is to access resume data. Another possibility is to use data from the Current Population Survey, which is a data product of the US Census Bureau. Comparing the edge weights with actual rates of workers transitioning between occupation pairs would strengthen the author's interpretation of the job network.

Thank you for pointing this out. I agree that validation of this similarity measure is crucial, and have utilized the CPS transition data in order to perform validation. New sections describing this validation procedure can be found in the main manuscript (see page 3, lines 64-69, and page 6, lines 184-193) and the supplementary material (ESM page 2). I have additionally added a supplemental figure (Figure S1) that visualizes this comparison both across the included jobs and within an example job.

Comment 2: Similar to (1), the author should validate their measures for connectivity between sectors against empirical data reflective of actual worker transitions. The author's use of job growth per sector offers an indirect validation, and I believe a more direct validation is possible.

I am unfortunately not fully clear on the distinction from comment 1. If this is in reference to the sector bridge coefficient, which is a composite of all of the inter-sector similarities within a job, its validity would follow directly from the validity of the individual similarities between job pairs.

However, the reference to job growth as an indirect validation leads me to believe that I am not fully understanding this suggestion. If it has not been addressed by the validation mentioned in response to comment 1, I would be happy to consider methods for performing this additional validation.

Comment 3: This study relies entirely on the automation estimates from the Frey & Osborne study, but this study is highly controversial. Therefore, the author should strengthen their claims by showing how their results change when different sets of estimates are considered. For this purpose, I recommend also considering

- O*NET Degree of Automation scores, and
- automation estimates from the Arntz et al study (ref. [5] in the article).

Thank you for raising this point, which is a very important one. I have indeed added new analyses to the paper that incorporate the O*Net degree of automation scores. However, I was unable to utilize them as a direct comparison to the F&O estimates, since their interpretation is quite different. Instead, I compared these two measures in the context of discussing the relatively weak relationship between jobs' current automation and their estimated future automation (see Figure 5).

In an attempt to obtain automation estimates that could be directly compared to the F&O estimates, I found many studies that created person-level or job-level estimates of automatability (including the Arntz study), as opposed to the occupation-level estimates presented by F&O. Since the proposed framework is designed for occupation-level estimates, I have added a section to the limitations section describing the potential benefit of accounting for individual- and job-level heterogeneity, and calling for future work to extend the network framework to allow for variability within occupations (see page 17, lines 554-663).

Comment 4: Although workers care about their jobs and policy makers care about employment numbers, work from Autor, Acemoglu, and others all suggest that occupations are best understood as abstract bundles of tasks and skills because technology interacts with labor by impacting the demand for specific tasks and skills. Therefore, does an occupation's susceptibility to automation impact the occupation's skill similarity to other occupations? In turn, does this fundamentally change the job network?

Work from James Bessen (BU) and, alternatively, from David Deming (Harvard) demonstrate that occupations that are exposed to automation can redefine their skill requirements to reflect changing labor demands. Often, these transitions focus on an increased reliance on social skills. Therefore, exposure to automating technology does not always reduce employment. As an example from Bessen's work, the increased adoption of ATMs corresponded to national increase in employment for bank tellers in the US--perhaps a counter-intuitive result--because ATMs made it cheaper to open bank branches (i.e., demand elasticity) and bank tellers began to replace clerical skills with social skills. So, since this occupational skill redefinition has been shown to occur with exposure to automating technology, how does the author incorporate these insights into our understanding of the job network in the study?

Thank you for bringing this to my attention. The idea of specific skills and tasks being affected by automation, as opposed to entire occupations, is quite important to consider in this context. In response to both this comment and the following comment (#5), I have added several new analyses designed to investigate how individual skills and knowledge areas relate to both current and future automation.

With respect to future automatability, I analyzed the links between individual skills and the F&O estimates of automatability, and described which specific skills tend to be associated with safer jobs, and how this varies across sectors (see page 12, lines 381-407, as well as page 16, lines 520-531).

Yet skills' associations with future automatability do not necessarily capture the variability within jobs that arises due to skill redefinition in the face of increased computerization. To address that, I utilized the O*Net degree of automation scores to detect skills that were important within jobs that had both (1) high rates of current automation and (2) low estimates of future automatability. Skills that showed this pattern can then be plausibly interpreted as skills that have become important within jobs that have already faced high levels of computerization and have adapted their tasks such that they are no longer susceptible to further automation. Skills highlighted in this analyses, therefore, may be candidates for retraining within jobs that will respond to automation by redefining tasks instead of decreasing employment (see pages 12-14, lines 408-440, as well as page 17, lines 532-552).

Comment 5: If the skill similarity scores that define the edge weights in the job network do indeed correspond to worker transitions between occupations, then which skills promote career mobility for workers? Are these the skills that are susceptible to automation (this may be difficult to answer convincingly)? More practically, this type of insight would be helpful to policy makers attempting to promote career mobility for workers through informed worker retraining programs. This level of insight should significantly strengthen the impact of this study.

See the response to comment 4 above, where I describe new analyses to investigate which individual skills may be the most beneficial for retraining, both in the context of retraining for job transitions, and in the context of retraining for within-occupation skill redefinition.

Reviewer 1 minor comments:

Comment 6: The "locally greedy" clustering algorithm employed in this study is often referred to as "Louvain community detection" in my experience.

Yes, you are absolutely correct. I have changed that wording on lines 196-197 to "a Louvain-like "locally greedy" algorithm" to be more specific (as the method used is not identical to the classical Louvain procedure), and I have added more detail in the Methods section regarding how this community detection procedure was utilized to obtain a consensus partition (see page 3, lines 70-84).

Comment 7: Figure 2 is very helpful for clarifying the measures.

Thank you, I was hopeful that this would be the case. I have additionally increased the size of the labels, and added boxplots within the violins, in an attempt to clarify the figure slightly.

Comment 8: On page 8, it is not clear to me what variable t is referring to. Is it the same t that is defined on page 9?

We have added more detail to the description of the transition value score, t , on page 5 (lines 131-143), and have moved extraneous details of the model to the ESM (see SI Methods).

Comment 9: Table 2 represents an excellent level of insight (of course, I have my concerns listed above). This is a level of detail that policy makers should find very helpful.

I appreciate this comment, and have designed similar tables for the new sections describing the benefit of individual skills (see Tables 2 and 3).

Reviewer 2

Reviewer 2 summary:

The study investigates the effects of automation on labor force and creates a network of jobs by their skills similarities. The identification of automation is based on the classification and the network results show distinct patterns of automation by different job sectors. In addition, the study introduces metrics for measuring job mobility. Finally, it offers a solution formula to estimate the cost/benefit of transitioning. Overall, the novel techniques and findings contribute to the field and provide novel solutions to workforce decision-makers. However, the paper suffers from a weaknesses in the methodology and analysis sections, mostly due to the lack of clarity and more detailed information.

I would like to thank the reviewer for her/his insights and recommendations. I believe that the resulting clarification and re-organization of the manuscript has greatly improved its value and potential impact, and very much appreciate the reviewer's time and effort.

Reviewer 2 comments:

Comment 1: Data section and methodology should be presented separately and prior to the analysis section.

Thank you for this comment. The data and methods descriptions have been expanded, and the paper has been re-arranged to place the full methods section prior to the results.

Comments 2-3: Please detail the size of BLS data, the number of occupations, the type of occupational code (e.g. SOC), how data was extracted (e.g. link to the BLS data) or mention that it is provided in supplemental materials.

It is not clear how BLS data, Frey's data, and ONET data are merged. Cite O*Net, BLS. Consider using the real job postings data with skills (see EMSI, BGT etc) in the future.

Thank you for these suggestions. Additional details regarding the nature and extraction of the BLS data, the size of the data, and the and the merging procedure have been placed in the “Data collection” section of the methods (see pages 2-3, lines 41-57).

Comment 4: Figure 1: How the number of clusters was determined? Medical cluster does not seem to be well-defined. It is not clear how sectors were assigned, for example using NAICS cross-walk?

I appreciate the opportunity to clarify. The network-based sectors were obtained using a community detection procedure that partitions networks into clusters based on the similarities between nodes. This procedure was repeated 100 times, and a consensus partition was developed. The number of clusters is not manually set, though it is closely tied to a parameter (λ) of the method. λ here was set at 1, which is a value commonly used for networks between 100 and 1000 nodes.

In addition to adding more detail on this procedure (see page 3, lines 70-84), I have also added a paragraph to the results describing the relationship between these network-based sectors and the CPS-defined occupation groups (page 7, lines 211-218). Additionally, I have added a supplemental figure (Figure S2) that visualizes the relationship between these two classification schemes.

Comment 4: Figure 1: Note also that in Frey's study, business was defined as low risk for automation, in Figure 1 business shows a bimodal distribution - perhaps two different categories?

This is a great point, and indeed it is the result of several of Frey & Osborne’s categories being consolidated into what I refer to as the “business” sector. The paragraph (page 7, lines 211-218) and supplemental figure (Figure S2) referenced in the previous response illustrate this in more detail.

Comment 5: Figure 2: The reviewer suggests to introduce briefly all 4 measures at the beginning of the section. Consider also adding labels to each figure, for example "local" or within and "distant" or inter

Thank you for this suggestion. I have added brief descriptions and formulas for the four mobility measures to the methods section. In case the metric labels in Figure 2 were too small or unclear, I have size them up and placed them more prominently between their respective graphs.

Comment 6: Figure 2: For Figure 2 E-H - Consider adding y-axis labels and median

I have added small boxplots within the violin plots for each measure (which include the medians as thin black horizontal lines). I did not add y-axis labels at this time, as the metrics don’t have inherent units. Thus, the y-axis labels would simply be the names of the metrics, which can currently be seen above the graphs. If the reviewer thinks there would be a clearer way to present this information, I would be happy to consider it.

Comment 7: Figure 2: If positive values in 2F indicate a bridge, then all 4 sectors are bridges [all positive]?

We appreciate the reviewer pointing this out. The previous phrasing that positive values represented bridges was incorrect. We have adjusted the description of this measure to reflect that it is bounded between 0 and 1, thus all nodes will necessarily have a positive value (see page 4, line 97)

Comment 8: Figure 2: 2G - not clear how business and industrial have lower bridge coefficients - is it based on their median?

This finding was based on pairwise t-tests, which were testing for differences in the mean values between sectors. While I agree that there is quite a bit of overlap between the values for the business/industrial sectors and those of the other three sectors, the test still revealed significant differences in their means. And though it is subtle, I think the figure does demonstrate that values tend to be lower in those sectors than in others (perhaps more-so now with the addition of the boxplots).

Comment 9: Could the bi-modality of business affect the inter-sector connectivity?

This is a very interesting question. The bimodality of automatability within the business sector would not have a direct effect on the connections of business jobs to jobs in other sectors. However, the fact that the business sector is made up of a more diverse set of jobs (according to the CPS classification groups; see Figure S2) may be related to its low inter-sector connectivity. Yet the service sector also contains a variety of CPS classifications and has relatively high inter-sector connectivity, so it is unclear to what extent this could drive the observed results.

Reviewer 2 minor comments:

Comment 10: Page 2 line 32 – spelling

Thank you, I have fixed this typesetting error.

Comment 11: Page 13 line 50 - spelling

Thank you, I have fixed this typesetting error.